# Structure–Activity Relationship of Novel ACE Inhibitory Undecapeptides from *Stropharia rugosoannulata* by Molecular Interactions and Activity Analyses

**DOI:** 10.3390/foods12183461

**Published:** 2023-09-17

**Authors:** Wen Li, Wanchao Chen, Jinbin Wang, Zhengpeng Li, Zhong Zhang, Di Wu, Mengqiu Yan, Haile Ma, Yan Yang

**Affiliations:** 1Institute of Edible Fungi, Shanghai Academy of Agricultural Sciences, Shanghai 201403, China; liwen3848@126.com (W.L.); chenwanchao@saas.sh.cn (W.C.); lizp_ln@126.com (Z.L.); zhangz0815@126.com (Z.Z.); vivian218074@163.com (D.W.); yanmengqiu@saas.sh.cn (M.Y.); 2School of Food & Biological Engineering, Jiangsu University, Zhenjiang 212013, China; 3Institute of Biotechnology Research, Shanghai Academy of Agricultural Sciences, Shanghai 201106, China; wangjinbin2013@126.com

**Keywords:** *Stropharia rugosoannulata* undecapeptides, molecular docking, molecular dynamics simulation, molecular thermodynamics reaction, molecular dynamics reaction, ACE inhibition mechanism

## Abstract

Undecapeptide is the central peptide molecule in the peptide base material of *Stropharia rugosoannulata*, and angiotensin-converting enzyme (ACE) plays a crucial role in hypertension. To fully explore the interaction mechanism and ACE-inhibitory activity of long-chain peptides from *Stropharia rugosoannulata*, the binding conformations of twenty-seven undecapeptides with the ACE receptor were revealed by molecule docking. The undecapeptide GQEDYDRLRPL with better receptor binding capacity and higher secondary mass spectral abundance was screened. All amino acid residues except proline in GQEDYDRLRPL interacted with the ACE receptor. GQEDYDRLRPL interfered with the receptor’s overall structure, with significant fluctuations in amino acid residues 340–355, including two residues in the receptor’s active pockets. The binding constants of GQEDYDRLRPL to the ACE receptors were at the μM level, with a kinetic binding constant of 9.26 × 10^−7^ M, which is a strong binding, and a thermodynamic binding constant of 3.06 × 10^−6^ M. Intermolecular interaction were exothermic, enthalpy-driven, and specific binding reactions. GQEDYDRLRPL had an IC_50_ value of 164.41 μmol/L in vitro and superior antihypertensive effects at low-gavage administration in vivo. Obtaining information on the interaction mechanism of ACE-inhibitory undecapeptides from *S. rugosoannulata* with the ACE receptor will help to develop and utilize ACE inhibitors of natural origin.

## 1. Introduction

Currently, the number of people with hypertension worldwide is constantly increasing. On 25 August 2021, a research paper titled “Worldwide trends in tension pressure and progress in treatment and control from 1990 to 2019”, coauthored by Imperial College London and the World Health Organization, was published in The Lancet [1]. Studies show that the number of adults aged 30 to 79 with hypertension has increased from 650 million to 1.28 billion in the last 30 years. Severe hypertension can cause physical problems like heart, brain, and kidney damage. Patients with malignant hypertension can develop kidney failure and even die quickly. The angiotensin-converting enzyme (ACE) plays a vital role in the induction of elevated blood pressure in vivo and is a drug target for the treatment of hypertension. ACE-inhibitory peptides can inhibit ACE activity and thus have a hypotensive effect. ACE-inhibitory peptides from food sources are considered natural substances and exhibit a high biological activity. They can be obtained by extraction, protein hydrolysis, and fermentation. Unlike antihypertensive drugs, food-derived ACE-inhibitory peptides have few side effects and a significant utilization value for people with hypertension and cardiovascular disease.

Peptides with ACE-inhibitory properties derived from food sources, whether plant or animal-based, are typically found in a mixture of peptide molecules. Isolating and purifying a single peptide fraction can be expensive, and the process is often limited to laboratory settings. Researchers commonly use LC-MS/MS techniques to analyze high-peptide-content mixtures. To predict peptide activity, functional peptide sequences are matched against the published literature. The actual activities of synthesized peptides are then validated through in vitro and in vivo tests. Notably, most peptide products currently available in the industry comprise a mixture of peptide molecules. Therefore, it is essential to determine the composition and conformational relationships of peptide molecules within mixed materials. This information can be used as a reference for the industrial applications of these peptide products.

*Stropharia rugosoannulata* is a grass rot fungus cultivated on a rice straw substrate. It is known for being rich in protein and peptides. The peptide products that are prepared by hydrolyzing the protein of *S. rugosoannulata* have been found to have excellent ACE-inhibitory activity (Patent No. CN202210297426.2). Despite this potential, the use of *S. rugosoannulata*’s ACE-inhibitory peptide in creating healthy food products has yet to be fully explored. The properties of peptides, such as their sequence, structure, and target receptor recognition, are closely linked to their active properties. Thus, creating a peptide library with different sequences is essential to explore their ACE inhibition mechanism further.

As long-chain peptides, polypeptides have a more complex spatial conformation than oligopeptides, resulting in more active peptide fragments that can be produced after digestion and hydrolysis. A library was constructed to investigate the potential ACE-inhibitory activity of undecapeptides (the main long-chain peptides identified by LC-MS/MS), and the toxicology and absorption activity of the undecapeptides were analyzed. Various techniques, including molecular docking, molecular dynamics simulation, molecular thermodynamics interaction, and molecular dynamics interaction, were employed to study the interaction between the undecapeptides and the ACE receptor. In vitro and in vivo analyses were carried out to verify the hypotensive function of the screened undecapeptide. This study could serve as a reference for developing highly active long-chain peptides.

## 2. Materials and Methods

### 2.1. Preparation of the Peptide Base Material of S. rugosoannulata

The *S. rugosoannulata* mushroom (mushroom strain NCBI No. SRR14469700) with a concentration of 48 g/L, 2 × 10^5^ U/g alkaline protease added amount of 1% (*w*/*w*) (Beijing Solarbio Science & Technology Co., Ltd., Beijing, China), hydrolysis temperature 42 °C, pH8.5 with synchronous ultrasonic-assisted enzyme hydrolysis conditions (bath ultrasonic power density 120 W/L, ultrasonic frequency 20 kHz, and hydrolysis time 40 min) were carried out to obtain the enzymatic hydrolysis solution of *S. rugosoannulata*. The hydrolysis supernatant was obtained by centrifuging the hydrolysis solution at 9000× *g* for 10 min at 4 °C and ultrafiltrate by the GE 3000 NMWC UF membrane. The permeate was freeze-dried at −70 °C for 48 h to obtain the peptide base material [2]. The method for identifying the peptide sequence of *S. rugosoannulata* was the same as that in the literature [3]. Peptide base materials were desalted, dissolved, centrifuged, and then taken for peptide sequence analysis by LC-MS/MS (Thermo-Obritrap-QE, ThermoFisher Scientific, Shanghai, China). The PEAKS 10.5 software (Bioinformatics Solutions Inc., Waterloo, ON, Canada) was used for peptide database retrieval.

### 2.2. Construction of S. rugosoannulata Undecapeptide Library with ACE Inhibitory Activity

The 82 undecapeptides identified by LC-MS/MS (Appendix A) were used for screening *S. rugosoannulata* undecapeptides with potential ACE-inhibitory activity. The peptide fragments with ACE-inhibitory activity in undecapeptides were identified by comparing the functional peptide sequence reported in the BIOPEP-UWM (https://biochemia.uwm.edu.pl/biopep-uwm/, accessed on 17 May 2023.). AdmetSAR (http://lmmd.ecust.edu.cn/admetsar1/predict/, accessed on 17 May 2023.) was used to predict the blood–brain barrier penetration, intestinal epithelial Caco-2 cell penetration, human intestinal absorption, and AMES oral toxicity of undecapeptides [4].

### 2.3. The ACE Inhibition Mechanism Assay of S. rugosoannulata Undecapeptides Based on Molecular Docking

Molecular docking is a computer simulation method to explain the binding patterns and interaction mechanisms between biomolecules at the atomic level. The crystal structure of the ACE receptor protein (1O8A) obtained from the PDB database was used as the target protein. MOE2019 (Chemical Computing Group ULC., Montreal, BC, Canada) was used to optimize the energy and structure of the ACE receptor and construct the 3D structure database of undecapeptides. Combining the critical amino acid residue (ALA354, GLU384, TYR523, GLN281, HIS353, HIS513, LYS511, TYR520, GLU162) in functional pockets (S1, S2, S1′) of the ACE receptor reported in the literature [5], the docking scoring value, the number of bonds formed, and the docking energy of undecapeptides to the ACE receptor were used as indicators to select the tightly bound undecapeptide–receptor complexes. Molecular docking was performed with 3 replications to verify the stability of the molecular simulation results. The Site Finder computed and applied the docking sites in the ACE receptor. The London dG score and number of poses were defaulted in MOE. MOE analyzed the interaction modes of the undecapeptides and the ACE receptor.

### 2.4. The ACE Inhibition Mechanism Assay of S. rugosoannulata Undecapeptides Based on Molecular Dynamics Simulation

Molecular dynamics (MD) simulation allows for dynamic analysis of the microscopic changes in intermolecular interactions at the atomic level. After a series of dynamic simulations, the intermolecular interactions system will reach the equilibrium state, and the analysis of the post-equilibrium trajectory can obtain kinetic data to predict the macroscopic properties of the interacting molecules. The MD simulation of undecapeptides and the ACE receptor was performed according to reference [6]. Amber99SB was chosen for the force field, and the TIP3P model was used for the water molecules. The receptor protein–undecapeptide complex system was first placed in the centre of a cubic box, and the distances between them were greater than 1.0 nm. The box was randomly filled with water molecules, and antagonistic sodium ions replaced the water molecules to make the simulated system electrically neutral. The simulated system was optimized using the energy minimization of the fastest decreasing method. Restricted MD simulation was used to equilibrate the systems. MD simulation was performed in an isothermal, isobaric system at a temperature of 300 K and a pressure of 1 atm for 50 ns. Molecular dynamics simulations were also conducted with 3 replications to verify the stability of the molecular simulation results. The conformation, the radius of gyration, and the formation bond changes between undecapeptide and the ACE receptor complex were calculated. MD simulation data analysis was performed by GROMACS 2022.3 software.

### 2.5. The ACE Inhibition Mechanism Assay of S. rugosoannulata Undecapeptide Based on Molecular Interactions

The interaction patterns between undecapeptide and the ACE receptor were analyzed using isothermal titration calorimetry (ITC) and biolayer interferometry (BLI) molecular interaction systems. The selected undecapeptide was synthesized by GL Biochem (Shanghai) Ltd., Shanghai, China.

The ACE receptor protein solution (0.02 mmol) prepared with PBS buffer was injected into the ITC cuvette at 25 °C. The operating program was as follows: stirrer speed, 1000 rpm; 60 s interval between start and first titration; 20 total injections; 2 μL (0.2 mmol) of buffer-prepared peptide solution in a single injection; 150 s interval between two titrations; single injection time, 2 s. Temperature changes in the reaction system were measured, and the ITC Nano analysis software 1.41 was used to calculate the complex thermodynamic binding affinity (K_D_), stoichiometry (N), enthalpy (ΔH), and entropy (ΔS) to verify whether the ACE receptor and undecapeptide interacted with each other.

The ACE receptor protein solution (1.0 mg/mL) prepared in PBS buffer was subjected to protein biotinylation (Genemore G-MM-IGT biotinylation kit, BMD Labservice Ltd., Suzhou, China). Protein biotinylation was incubated for 30–60 min and desalted. Four SSA sensors were used to solidify biotinylated proteins, and 4 SSA sensors were used as reference electrodes. Set the baseline equilibration time, loading sample time, and baseline equilibration time of 60 s–800 s–60 s. The peptide solutions were prepared at concentrations of 116.6 μmol/L, 233.2 μmol/L, 466.4 μmol/L, 932.8 μmol/L, 1866 μmol/L, and 3731 μmol/L. A biomolecular interaction cycle program with a baseline equilibration time of 60 s, a binding time of 120 s, and a dissociation time of 120 s was established to perform the interactions between the ACE receptor and undecapeptide at 25 °C. The complex’s binding constant (K_on_), dissociation constant (K_off_), and kinetic affinity K_D_ were calculated to determine the interaction modes between the ACE receptor and undecapeptide.

### 2.6. The ACE Inhibitory Activity In Vitro and Hypotensive In Vivo Assay of S. rugosoannulata Undecapeptide

The 0.1 g synthesized peptide was weighed and dissolved in 50 mL of pure water. The solution was diluted step by step to obtain the undecapeptide solutions with concentration gradients of 2.0, 0.4, 0.08, 0.016, and 0.0032 mg/mL. The ACE-inhibitory activity of the undecapeptide solutions was measured by the DOJINDO ACE Kit-WST kit (Shanghai Youlu Biotechnology Co., Ltd., Shanghai, China). The ACE inhibition activity was calculated by detecting the absorbance at 450 nm of 3-Hydroxybutyric acid (3HB), which was produced by ACE-catalyzed 3-Hydroxybutyryl-Gly-Gly-Gly (3HB-GGG). The ACE inhibition activity curve was made with the sample concentration and the inhibition rate, and the sample concentration at 50% inhibition rate (IC_50_) was calculated.

Eleven-week-old male SHR rats and Wistar rats (SiPeiFu (Suzhou) Biotechnology Co., Ltd., Suzhou, China) weighing 200–250 g were used. After 7 d of acclimatization feeding, SHR rats and Wistar rats were randomly grouped into 6 rats per group for a single gavage administration treatment. The undecapeptide and the commercial antihypertensive drug Benazepril hydrochloride were administered in a gavage dose of 10 mg/kg body weight (the recommended dose of Benazepril hydrochloride). SHR rats with regular saline treatment were used as the blank control group. The systolic blood pressure (SBP) and diastolic blood pressure (DBP) of rats were monitored by the Softron BP system (Softron BP-2010A, Tokyo, Japan) at 0, 2, 4, 6, 8, and 24 h after drug administration. At the end of the treatment, the organ coefficients were calculated (organ weight as a percentage of the body weight of the rat, mg/g). The animal study was reviewed and approved by the Yanxuan Biotechnology Laboratory Animal Welfare & Ethical Committee (Hangzhou) Co., Ltd., Hangzhou, China (YXSW 2201184105).

### 2.7. Statistical Analysis of the Data

The Origin 2019 software was used to plot the variation of the MD simulation parameters. GraphPad Prism 9 software was used for thermodynamics and dynamics data plots obtained from ITC and BLI analysis. The ACE-inhibitory activity in vitro experiment was performed thrice, and the 50% inhibition rate data was calculated. Data from the in vivo tests were presented as means ± standard deviations of 6 measurements.

## 3. Results

### 3.1. Undecapeptides with ACE Inhibitory Activity in S. rugosoannulata

A peptide database of 27 undecapeptides with potential ACE-inhibitory activity was constructed. The active ACE-inhibitory peptide fragments were plentiful, with the main ones being PL, RP, LP, and PG, which had high frequencies. The prediction results of admetSAR showed that all 27 undecapeptides were non-toxic (0.667–0.900). The blood–brain barrier penetration (0.824–0.997), Caco-2 cell penetration (0.702–0.875), and human intestinal absorption properties (0.518–0.964) of undecapeptides were good (Appendix A).

### 3.2. Molecular Docking Results of the ACE Receptor and Undecapeptides

MOE was used to analyze the active amino acid residues in the ACE receptor (Figure 1A, 133 amino acid residues). The spatial conformation and docking sites of 27 undecapeptides bound to the ACE receptor are shown in Figure 1B–F and Appendix A. It can be seen that the undecapeptide conformations were more stretching in the receptor cavity, and the distribution positions were extensive (the orange dashed box of Figure 1B–F). There were 56 types of amino acid residues in the receptor to which the undecapeptide could bind, and hydrogen bonds and ionic bonds were the central interaction bonds in undecapeptides and the ACE receptor complex (Appendix A). The amino acid residues (ALA354, GLU384, TYR523, GLN281, HIS353, HIS513, LYS511, TYR520, GLU162) in the functional pockets (S1, S2, and S1′) of the ACE receptors were used as indicators, and the interaction types and binding energy of undecapeptides are shown in Appendix A. Except for the residue TYR520 in the active pocket S2, the 27 undecapeptides could form interactions with the other eight critical amino acid residues in varying amounts. The binding energy range was −21.3~−0.5 kcal/mol (the lower the energy of the interaction bonds, the more stable the complex formed). The amino acid residues ALA354, GLU384, HIS353, HIS513, and GLU162 in the receptor formed interactions with more undecapeptides (10–12 undecapeptides), among which GLU384 and GLU162 formed more binding bonds with undecapeptides (19 and 27 bonds, respectively). The bond number formed by LYS511 with undecapeptides was lower than the above two amino acid residues. Still, its total binding energy was lower (−91.6 kcal/mol), second only to that formed by undecapeptide with GLU162 (−99.1 kcal/mol).

The 27 undecapeptides also formed many binding bonds with other amino acid residues in the structural domain of the receptor. LYS118, GLU403, ARG522, and GLU123 were the four primary amino acid residues that formed interactions with undecapeptides. The number of bonds formed by undecapeptides with the four amino acid residues was more than 27, and the total binding energy was −156.7 kcal/mol~−71.7 kcal/mol. It has been reported that HIS383, HIS387, GLU411, and Zn^2+^ were the ACE receptors’ catalytic sites and ACE-inhibitory peptides’ primary binding sites [5,7]. Although the bond number formed by GLU411 and HIS387 with undecapeptides was low (10 and 11, respectively), the total binding force created by the above two amino acid residues with undecapeptides was strong (binding energy −78.9 kcal/mol and −70.2 kcal/mol, respectively), which contributed a lot to maintaining the stability of undecapeptides and the ACE receptor complex. In addition, undecapeptides also formed many low-energy metal bond/ion forces with zinc ions. The abundance of amino acid residues and zinc ions bound to undecapeptides and the strong interaction force played an essential role in the ACE-inhibitory activity of undecapeptides.

Among the 27 undecapeptides, GQEDYDRLRPL had the highest MS abundance (6.80 × 10^7^), and the peptide contained the ACE-inhibitory peptide fragments RL, LRP, PL, RP, GQ, DY, LR, and DR. The peptide bound firmly to critical amino acid residues GLU384, HIS513, LYS511, and GLU162 in S1, S2, and S1′ pockets, forming strong hydrogen bonds and electrostatic interaction forces (−19.8 kcal/mol binding energy with GLU162; −20.5 kcal/mol binding energy with LYS511). The peptide formed low-energy bonds with the critical amino acid residues (LYS118, GLU403, ARG522) and ASP358. The binding energy was −6.4 kcal/mol with ARG522, −9.1 kcal/mol with LYS118, −10.7 kcal/mol with GLU403, and −11.9 kcal/mol with ASP358. The docking amino acid residues from GQEDYDRLRPL mainly included amino nitrogen atoms (G1, Q2, Y5, R7, R9, L11), carbonyl oxygen atoms (D4, Y5, L8), and carboxyl oxygen atoms (E3, D6, L11) (Figure 2). In summary, GQEDYDRLRPL could interact tightly with critical amino acid residues in S1, S2, and S1′ and other key amino acid residues in the receptor structural domain. Except for proline (P), all amino acid residues in GQEDYDRLRPL exerted interactions with the ACE receptor. It was hypothesized that GQEDYDRLRPL might have better ACE-inhibitory activity. Therefore, GQEDYDRLRPL was selected for MD simulation and molecular interactions.

### 3.3. Molecular Dynamics Simulation Results of GQEDYDRLRPL and the ACE Receptor

The results of the MD simulation of GQEDYDRLRPL with the ACE receptor are shown in Figure 3. The root mean square deviation (RMSD) was used to measure the overall change in the structural conformation of the complex during MD simulation and to observe and evaluate the protein stability. The MD simulation system had a more pronounced effect of GQEDYDRLRPL on the ACE receptor between 0 and 15 ns, and the RMSD fluctuated widely, which interfered with the overall structure of the receptor. The RMSD of the system stabilized after 15 ns, and the complex stabilized around 2.5 Å. (Figure 3A).

The root mean square fluctuation (RMSF) is the deviation of individual amino acid residues from their average position in the receptor over time. It is used to analyze the protein structural fluctuation region. Amino acid residues with high RMSF values have more extraordinary mutability and are the central structural region where peptides affect the receptor. The interaction between GQEDYDRLRPL and the ACE receptor significantly affected amino acid residues 340–355 region. The RMSF in this region was significantly higher than those in other positions, indicating that it was the hot spot region for GQEDYDRLRPL to affect the ACE receptor (Figure 3B), and amino acids fluctuations in this region would affect the state structure of the receptor protein in solution, interfering with protein–peptide interactions. Amino acid residues in this region include MET340, LEU341, GLU342, LYS343, PRO344, THR345, ASP346, GLY347, ARG348, GLU349, VAL350, VAL351, CYS352, HIS353 (in the S2 pocket), ALA354 (in the S1 pocket) and SER355. The molecular docking results showed that GQEDYDRLRPL did not bind to the amino acid residues mentioned above in the final steady-state complex to generate interaction forces. Thus, it is presumed that the amino acid residues bound and dissociated from the GQEDYDRLRPL during the MD simulation, leading to the perturbing of the amino acid residues. This region was also the central region of structural changes in the receptor. Furthermore, the fluctuation of other active amino acid residues with which the peptide interacted was less, indicating that GQEDYDRLRPL and the ACE receptor binding complex had good stability.

The radius of gyration is used to assess the structural compactness of the protein–peptide complex in the MD simulation system, and a smaller radius of gyration indicates a dense structure. The radius of gyration changes during the MD simulation tended to decrease (Figure 3C), favouring the interaction of GQEDYDRLRPL with the ACE receptor. During the 0–50 ns MD simulation process, there were no significant changes in the ACE receptor’s solvent-accessible area (Figure 3D) and the secondary structure (Figure 3E). The number of hydrogen bonds within the protein–peptide complex decreased slightly during the 10–15 ns process, which may be related to the influence of GQEDYDRLRPL on the receptor structure. After 15 ns, the number of hydrogen bonds gradually increased and levelled, indicating that the binding of the peptide to the receptor tended to have a stable structure (Figure 3F).

### 3.4. Molecular Thermodynamics and Dynamics Interactions Results of GQEDYDRLRPL and the ACE Receptors

The ITC interaction results showed that the enthalpy (−440 KJ/mol) and the entropy (−1408 J/mol·K) between GQEDYDRLRPL and the ACE receptor interaction were negative (Gibbs free energy < 0, enthalpy-driven reaction) and the reaction system was spontaneously exothermic. The stable complex was formed by GQEDYDRLRPL interacting with multiple sites on the receptor (N = 2.22). Hydrogen bond interaction and van der Waals forces were the main driving forces for reciprocal binding. The thermodynamic binding affinity K_D_ value of the receptor and GQEDYDRLRPL was 3.06 × 10^−6^ M (Figure 4A,B). The ITC analysis results were consistent with the molecular docking results.

The BLI dynamics interaction results of GQEDYDRLRPL with the ACE receptor at different gradient concentrations showed that the binding constant K_on_ of GQEDYDRLRPL to the ACE receptor was 6.84 × 10^5^ (M^−1^ S^−1^). The primary rate of peptide binding to the receptor was fast, and after 120 s, GQEDYDRLRPL binding to the receptor reached a saturation stage, and the interaction tended to equilibrium. The signal response increased dose-dependently with peptide concentration, and the peptide activity bound to the receptor was better at medium to high concentrations (1866 μmol/L and 3731 μmol/L). The kinetic affinity value K_D_ of GQEDYDRLRPL to the ACE receptor was 9.26 × 10^−7^ M, a high intensity binding level in protein–peptide molecule interactions. GQEDYDRLRPL was explicitly bound to the ACE receptor (Figure 5). The dynamic results of the BLI analysis were consistent with the ITC thermodynamic results.

### 3.5. The ACE Inhibitory Activity In Vitro and Hypotensive Evaluation In Vivo of GQEDYDRLRPL

The in vitro ACE inhibition curve of GQEDYDRLRPL obtained by the kit method was y = 8.4157 ln(x) + 62.597 (R² = 0.903), and its IC_50_ value was 164.41 μmol/L. The results of SBP and DBP in spontaneously hypertensive rats (SHR) showed that SHR rats treated with GQEDYDRLRPL and Benazepril hydrochloride by gavage had better blood pressure lowering effects after 2, 4, 6, and 8 h. The most significant blood pressure lowering effect was observed at 8 h. Blood pressure values of the treated groups began to rebound after 8 h. In the GQEDYDRLRPL treatment group, the decrease in SBP was 37.98 ± 1.26 mmHg, and the reduction in DBP was 29.64 ± 1.23 mmHg. In the Benazepril hydrochloride treatment group, the reduction in SBP was 54.94 ± 1.79 mmHg, and the decrease in DBP was 39.17 ± 1.81 mmHg. In the GQEDYDRLRPL gavage-treated group of Wistar rats, the drop in SBP and DBP was 9.11 ± 0.85 mmHg and 2.44 ± 1.20 mmHg, respectively. The effect of GQEDYDRLRPL on the blood pressure of Wistar rats was insignificant. (Figure 6A,B).

The heart organ coefficients in SHR rats showed a significant difference between the Benazepril-hydrochloride-treated group and the blank control group, and there was no significant difference between the GQEDYDRLRPL-treated group and the blank control group. The kidney organ coefficients were significantly lower in the Benazepril hydrochloride treatment group of SHR rats and considerably higher in the GQEDYDRLRPL treatment group of Wistar rats (Figure 6C,D). Benazepril hydrochloride treatment had a more significant effect on the SHR rats’ organs. The low body weight in two Wistar rats of the GQEDYDRLRPL gavage-treated groups might be the main reason for the high coefficients of the kidney. In conclusion, GQEDYDRLRPL had a hypotensive effect on SHR rats by gavage treatment and had less impact on the organs of SHR rats. The dosage used for therapy has been lowered to serve as a raw material for the production of hypotensive functional foods.

## 4. Discussion

Studies have concluded that elevated blood pressure can put patients at risk for memory loss and Alzheimer’s disease [8]. Data from the Blood Pressure Lowering Treatment Trials’ Collaboration (BPLTTC) meta-analysis extend findings from an earlier meta-analysis that showed the cardiovascular benefits of blood-pressure-lowering drugs in patients. Over an average of 4 years of follow-up, each 5 mmHg reduction in systolic blood pressure lowered the relative risk of major cardiovascular events by approximately 10% (https://www.medscape.com/viewarticle/936598, accessed on 17 May 2023.). It shows that hypertensive patients need to reduce their blood pressure, and blood pressure regulation in ordinary people also significantly reduces the risk of cardiovascular events. Therefore, active peptide products with blood-pressure-lowering functions will have a vast application space. Using mushrooms to prepare natural bioactive peptides can enhance the value of mushrooms and reduce the production cost of peptides.

Peptide profiling identification [7,9,10,11], toxicological prediction [12,13], and molecular docking [7,9,11,12,13,14,15,16,17] are effective means to obtain peptides with high ACE-inhibitory activity in vitro rapidly. Researchers have reported a variety of food-derived ACE-inhibitory peptides from animal protein sources such as dairy products [9] and seafood [7,12,16,18,19], as well as from soy products [14,20], high-quality crops or low-value plant protein [15,21,22,23,24]. The varied sequences of amino acids in peptides have increased the diversity of food-derived peptides. Food-derived ACE-inhibitory peptides were mainly water-soluble and non-toxic. Their inhibited mechanism often includes mixed or competitive mode, binding to ACE’s active or inactive sites through hydrogen bonds, electrostatic interactions, and hydrophobic interactions [12,13,14,15,16,25]. The 27 undecapeptides could inhibit ACE activity in a mixed mode by forming hydrogen bonds and electrostatic interaction forces with amino acid residues in the active and inactive sites of the ACE receptor.

MD simulation [18,26], molecular interactions [18,27], simulated digestion in vitro [9,10,11,20,25,28], and antihypertensive effects evaluation in vivo [14,22,29] effectively verify and reveal the ACE inhibition mechanism. The peptide–receptor complex’s interaction mode and conformation changes can help improve our understanding of the inhibition mechanism. Evaluating the activity of the ACE-inhibitory peptide after digestion and absorption can obtain information on the peptide’s stability and activation pathway. ACE-inhibitory peptides are usually time-dependent and dose-dependent in regulating blood pressure. The regulatory path involves the renin–angiotensin system (RAS), amelioration of oxidative stress-induced cell damage, activity activation of nitric oxide synthase and gene expression in endothelial cells, and regulation of nitric oxide (NO) and endothelin-1 (ET-1) production [19,22,30,31,32,33,34]. The cross-validation of multiple techniques laid the theoretical foundation for revealing the interaction mechanism of ACE-inhibitory peptide and its hypotensive effects. Molecular docking and molecular dynamics simulations cross-validated that GQEDYDRLRPL can form a tightly structured complex (2.5 Å) with the ACE receptor, with hydrogen bonding interaction driving the peptide molecule to bind to the receptor molecule at multiple sites. Cross-validation of the dynamics and thermodynamic results of the intermolecular interactions confirmed the existence of a moderate affinity force between the biomolecules, and cross-analysis of the in vitro ACE-inhibitory activity and in vivo hypotensive activity confirmed that GQEDYDRLRPL is a novel antihypertensive agent.

The biological activity of peptides is correlated with the type of amino acids in the sequence and its secondary structure. Generally, peptides containing hydrophobic and aromatic amino acid residues at the N- and C-terminus, proline at the second C-terminus position, or those containing arginine and proline had better ACE-inhibitory activity [16,35,36,37]. Peptides containing negatively charged amino acids were essential for their interaction with ACE and their ACE-inhibitory activity [37]. GQEDYDRLRPL contained hydrophobic amino acid leucine at the C-terminus, arginine, and proline in the peptide sequence and had a negative charge at pH 7.0, which is consistent with the above amino acid composition patterns of the ACE-inhibitory peptides. However, since GQEDYDRLRPL is a long-chain peptide, the spatial position of proline in its secondary structure affects its binding to ACE receptors, as shown by the molecular docking results. Therefore, proline in GQEDYDRLRPL did not directly contribute to the ACE-inhibitory activity of GQEDYDRLRPL.

There is some difference between the ACE-inhibitory activity in vitro and the hypotensive activity in vivo. Peptides with relatively poor ACE-inhibitory activity exhibited better hypotensive activity after digestion in vivo, possibly related to the peptide degradation fragments with a higher ACE-inhibitory activity. GQEDYDRLRPL had a hypotensive effect at lower gavage treatment doses. It is reported that PL was an ACE-inhibitory peptide with an IC_50_ value of 337.3 muM [38]. Virtual digestion of GQEDYDRLRPL by gastrointestinal protease could produce the peptide fragments of GQEDYDR, L, R, and PL, and it is hypothesized that PL peptide fragments also played a role in blood pressure regulation in SHR rats.

## 5. Conclusions

PL, RP, LP, and PG were the main peptide fragments with ACE-inhibitory activity among the *S. rugosoannulata* undecapeptides. Undecapeptides stretched conformationally in the ACE receptor’s active cavity, forming low-energy bonds with the receptor’s amino acid residues and zinc ions. The screened undecapeptide, GQEDYDRLRPL, except for proline (P) in the peptide, could bind tightly to the ACE receptor. GQEDYDRLRPL interfered with the overall structure of the ACE receptor, with large fluctuations in the amino acid residues 340–355 region. After 15 ns, the binding complex tended to have a stable structure. The interaction between GQEDYDRLRPL and the ACE receptor was an enthalpy-driven spontaneous exothermic reaction with specific binding. GQEDYDRLRPL had an ACE-inhibitory value (IC_50_) of 164.41 μmol/L in vitro. At a therapeutic dose of 10 mg/kg body weight by gavage in SHR rats, a significant reduction in blood pressure was observed within 8 h. This study provided a theoretical basis for the *S. rugosoannulata* undecapeptides with ACE-inhibitory activity for producing healthy foods.

## Figures and Tables

**Figure 1 foods-12-03461-f001:**
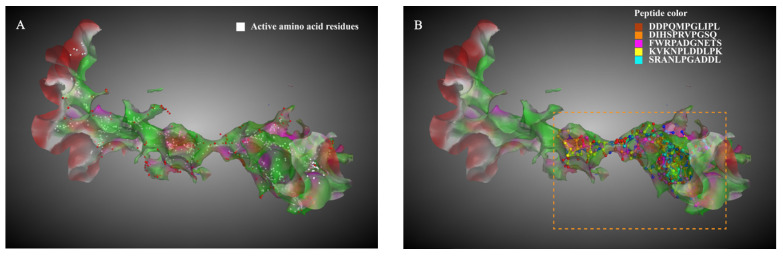
Spatial conformation of the 27 *S. rugosoannulata* undecapeptides bound to the ACE receptor. (**A**) The active cavity of the ACE receptor and the active amino acid residues (white sphere); (**B**) main undecapeptides bound to the amino acid residue ALA354; (**C**) main undecapeptides bound to the amino acid residue GLU384; (**D**) main undecapeptides bound to the amino acid residue HIS353; (**E**) main undecapeptides bound to the amino acid residue LYS511; (**F**) main undecapeptides bound to the amino acid residue GLU162. The orange dashed box showed the interaction region of undecapeptides and the ACE receptor.

**Figure 2 foods-12-03461-f002:**
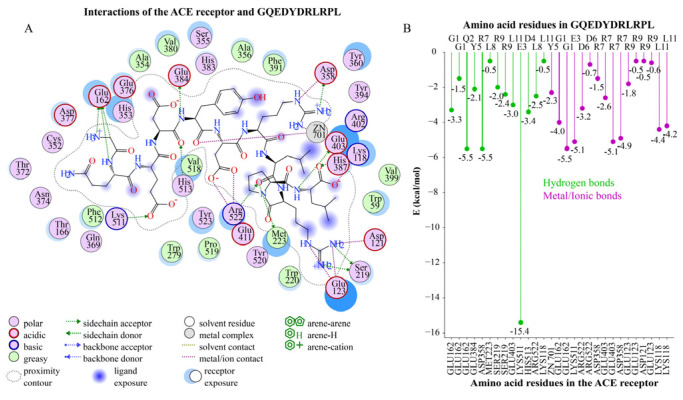
The 2D docking plot of GQEDYDRLRPL and the ACE receptor. (**A**) Docking residues; (**B**) docking bond energy.

**Figure 3 foods-12-03461-f003:**
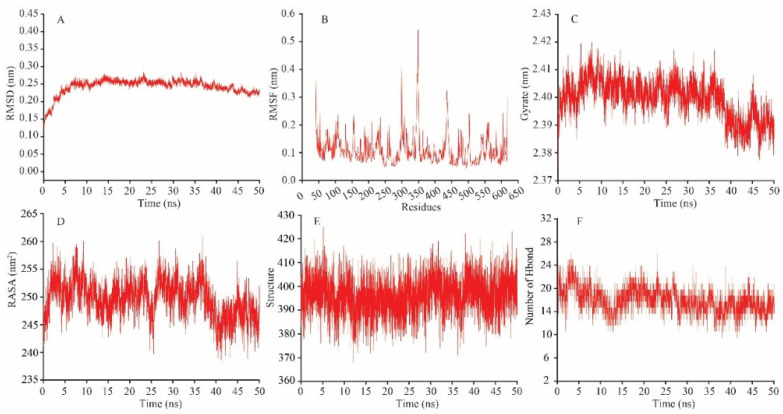
Molecular dynamics simulation results. (**A**) RMSD changes during MD simulation; (**B**) RMSF changes during MD simulation; (**C**) the radius of gyration changes during MD simulation; (**D**) the receptor’s solvent-accessible area changes during MD simulation; (**E**) the receptor’s secondary structure changes during MD simulation; (**F**) the number of hydrogen bond changes during MD simulation.

**Figure 4 foods-12-03461-f004:**
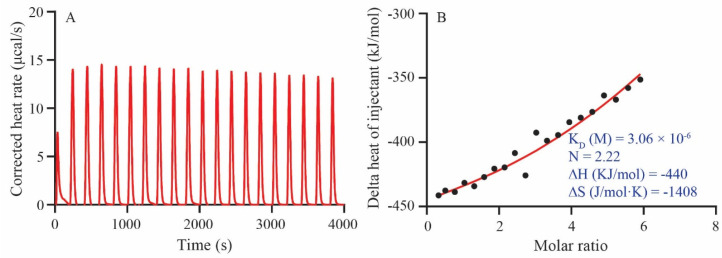
Molecular thermodynamics interactions between GQEDYDRLRPL and the ACE receptor. (**A**) Pulse diagram of heat generated per titration; (**B**) molar heat to molar ratio heat integral fitting plot.

**Figure 5 foods-12-03461-f005:**
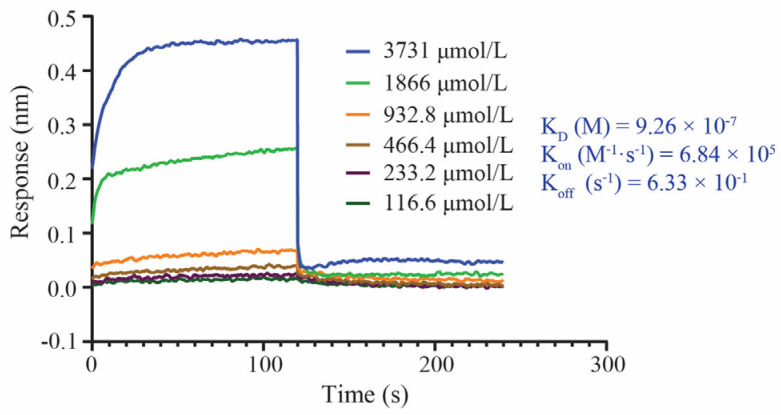
Molecular dynamics interactions between GQEDYDRLRPL and the ACE receptor.

**Figure 6 foods-12-03461-f006:**
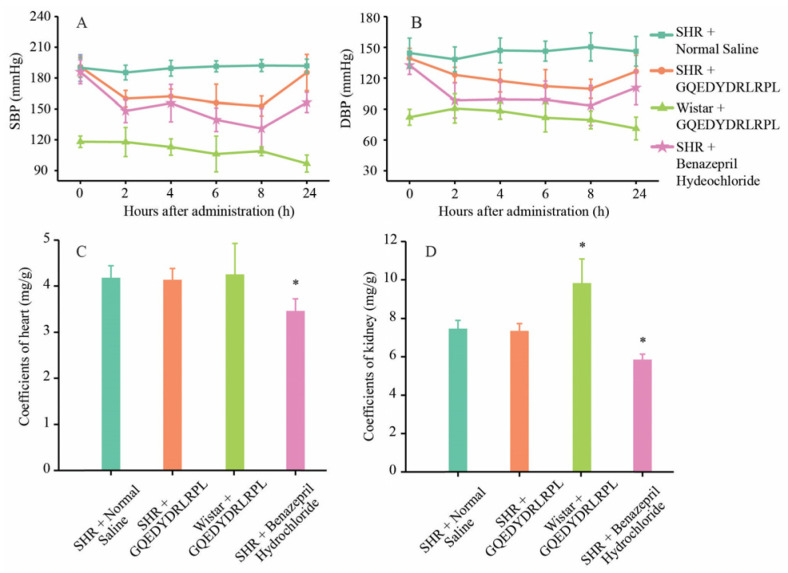
Blood pressure values and organ coefficients of rats at different periods after drug administration. (**A**) Systolic blood pressure; (**B**) diastolic blood pressure; (**C**) heart coefficients; (**D**) kidney coefficients. * indicates a significant difference compared to the blank control group (*p* < 0.05).

## Data Availability

The data used to support the findings of this study can be made available by the corresponding author upon request.

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
