# Peer review of "Structure–Activity Relationship of Novel ACE Inhibitory Undecapeptides from Stropharia rugosoannulata by Molecular Interactions and Activity Analyses"

_foods, 2023, doi:10.3390/foods12183461_

Round 1

Reviewer 1 Report

In this study, the authors examined the anti-ACE potential of 27 undecapeptides identified from Stropharia rugosoannulata and confirmed their prediction (focusing on GQEDYDRLRPL) by means of in vitro and in vivo assays. Overall, the study was interesting, providing valuable information. However, certain sections, especially M&M, appear unclear and may require further checks/amendments.

Below are a few minor feedbacks for the authors’ consideration:

1.     It is unclear to me why the authors only focused on undecapeptides. It seems unlikely that their hydrolysis would only release 11-residue peptides and that no peptides of other lengths were liberated. Moreover, the longer the peptides, the more susceptible they would be to breakdown by gastrointestinal digestion. And if the latter indeed happens, the actual (final) peptide sequences exerting the anti-ACE or anti-hypertensive effect would not be the initial 11-residue sequence reported in the manuscript. So, it would be good if the authors can explain their decision of focusing on undecapeptides and the risk of susceptibility to gastrointestinal digestion more explicitly this in the text.

2.     I am aware that the authors have performed in silico screening to evaluate the potential gastrointestinal absorption of their peptides. However, gastrointestinal absorbability cannot be regarded as equivalent to resistance to degradation by the gastrointestinal proteases.

3.     In the abstract, more specific information should be provided, e.g., the part about antihypertensive effects (the last statement). In addition, the rationale for focusing on 11-residue peptides should be briefly stated in the abstract.

4.     In M&M:

a)     It is unclear whether the enzymatic hydrolysis parameters/conditions employed by the authors (lines 81-86) were based on those previously reported by others. This info is important as it may influence the pool of peptides liberated from the sample and subsequently identified by the authors. Please clarify.

b)    Line 96 (…data link, http://139.224.23.107) – It seems not a good idea to store the info about the 82 undecapeptides on that webpage. When I tried to access it, I found that it requires sign-up and log-in. Also, the webpage is in the Chinese language, which makes it inaccessible to readers who do not understand Chinese. Considering the open access nature of this journal, it is preferable to make the info more accessible to everyone by putting it in the supplementary data files.

c)     For molecular docking, the authors used the crystal 1O8A, which is a testes isoform of ACE. Why didn’t the authors use the crystal for the somatic ACE instead, e.g., the crystal 4APJ? After all, the somatic ACE is more broadly distributed in the body when compared with the testis ACE which is restricted to the sperms.

d)    In lines 111-113, the authors mentioned some “critical amino acid residues”. Please cite references which report that those amino acid residues are critical.

e)     For molecular docking and dynamics, did the authors perform three replicates too, or just one replicate? Was redocking performed to validate that the docking parameters adopted were reliable? Also, what were the docking parameters used? The information appears not indicated in the manuscript. Please check.

f)     For molecular dynamics simulation, 50 ns is rather short. Did the authors attempt to simulate for up to 100 ns, and perform replicates too?

5.     For prediction of anti-ACE potential, the authors searched the BIOPEP-UWM database to see whether their undecapeptides contain previously reported short anti-ACE sequences. While this is often done by others as reported in the literature, it may also restrict the discovery of novel anti-ACE sequences. The authors can consider using various free online anti-ACE peptide prediction servers to screen their sequences instead.

6.     Did the authors compare the anti-ACE activity of GQEDYDRLRPL with any well-established anti-ACE agent or antihypertensive drugs?

7.     Discussion can be improved. The first three paragraphs sound like literature and hardly discuss any data/results obtained in this study.

8.     In the section 2.7 (lines 187 – 189), it seems inappropriate to describe MOE 2019 and MD simulation software here as these are not considered to be statistical software.

Author Response

Dear Editors and Reviewers:

Thank you for your letter and for the reviewers' comments concerning our manuscript entitled "Structure-activity relationship of novel ACE inhibitory un-decapeptides from Stropharia rugosoannulata by molecular interactions and activity analyses" (Manuscript ID: foods-2617741). We have studied the comments carefully and have made corrections which we hope meet with approval. Revised portions are marked in red in the manuscript. The major revisions in the paper and the responses to the reviewer's comments are as follows:

Reviewer #1:

Comments and Suggestions for Authors

In this study, the authors examined the anti-ACE potential of 27 undecapeptides identified from Stropharia rugosoannulata and confirmed their prediction (focusing on GQEDYDRLRPL) by means of in vitro and in vivo assays. Overall, the study was interesting, providing valuable information. However, certain sections, especially M&M, appear unclear and may require further checks/amendments.

Below are a few minor feedbacks for the authors’ consideration:

1. It is unclear to me why the authors only focused on undecapeptides. It seems unlikely that their hydrolysis would only release 11-residue peptides and that no peptides of other lengths were liberated. Moreover, the longer the peptides, the more susceptible they would be to breakdown by gastrointestinal digestion. And if the latter indeed happens, the actual (final) peptide sequences exerting the anti-ACE or anti-hypertensive effect would not be the initial 11-residue sequence reported in the manuscript. So, it would be good if the authors can explain their decision of focusing on undecapeptides and the risk of susceptibility to gastrointestinal digestion more explicitly this in the text.

Thanks for the reviewer's comments.

Hydrolyzed peptide base materials contain undecapeptide and other length peptide molecules. The reason why we focused on undecapeptides in this study was mainly from the following considerations:

(1) Undecapeptides accounted for more hydrolyzed peptide matrices and were the main peptide component [Ultrasonics Sonochemistry, 2022, 90, 106206].

(2) Based on the comparison of molecular docking results between peptides and oligopeptides, we found that the long-chain peptides (undecapeptides) have a more significant binding advantage in receptor binding, compared to the oligopeptides (number of residues less than 10), have better receptor binding properties, and all 27 undecapeptides in this study can bind to amino acid residues in the ACE receptor active pockets S1, S2 and S1'. Therefore, we conducted in vivo and in vitro studies on the hypotensive activity of the undecapeptides and oligopeptides in the peptide base material, respectively (the results of the oligopeptides have not been published yet).

(3) Long-chain peptides undergo gastrointestinal digestion to produce more peptide fragments, which may have synergistic effects in antihypertension, and we expect that both the selected peptide molecules and their degradation products will have specific antihypertensive properties.

(4) We have also conducted gastrointestinal digestion studies of peptide base mixtures (results not yet published), mainly considering the application of peptide base mixtures in health foods. The application of a single peptide molecule is more considered to be applied in functional products for blood pressure regulation, and its gastrointestinal digestion properties are evaluated by virtual simulation of gastrointestinal digestion, with digested peptide fragments GQEDYDR, L, R, and PL, which is informative to understand the digestive properties of peptide molecules.

(5) From the results of the in vivo antihypertensive effect evaluation in SHR rats, it can be seen that the blood pressure continued to decrease within 8 h after the GQEDYDRLRPL gavage treatment, which indicates that the peptide molecule (or its degraded fragments) has some anti-digestive properties. Whether the in vivo antihypertensive treatment is still the peptide molecule initially administered, this work needs to be further explored using molecular labeling and other methods. If an easily digestible situation exists, peptide molecule protection using molecular embedding and other means will follow. The recommendations of the reviewer deserve further improvement in our future work.

2. I am aware that the authors have performed in silico screening to evaluate the potential gastrointestinal absorption of their peptides. However, gastrointestinal absorbability cannot be regarded as equivalent to resistance to degradation by the gastrointestinal proteases.

Thanks for the reviewer's comments.

Indeed, the gastrointestinal absorbability of a peptide molecule cannot be equated with its resistance to degradation by gastrointestinal proteases. We are evaluating the resistance of peptide molecules to GI protease degradation by virtual gastrointestinal digestion simulations. This part of the data was not presented in the previously submitted manuscript, and we have now added this part of the results to Supplementary Table S2 (Original Supplementary Table S1).

3. In the abstract, more specific information should be provided, e.g., the part about antihypertensive effects (the last statement). In addition, the rationale for focusing on 11-residue peptides should be briefly stated in the abstract.

Thanks for the reviewer's comments. We have added the relevant contents, and please check the revised parts of lines 17-18 and 30-32.

4. In M&M:

a) It is unclear whether the enzymatic hydrolysis parameters/conditions employed by the authors (lines 81-86) were based on those previously reported by others. This info is important as it may influence the pool of peptides liberated from the sample and subsequently identified by the authors. Please clarify.

Thanks for the reviewer's comments. Our previous research optimized the enzyme hydrolysis conditions used in the study, and the research report on the optimized conditions has been published in Chinese (https://doi.org/10.16429/j.1009-7848.2023.08.024.). We have added the information of the published article here in the form of a reference citation.

b) Line 96 (data link, http://139.224.23.107) – It seems not a good idea to store the info about the 82 undecapeptides on that webpage. When I tried to access it, I found that it requires sign-up and log-in. Also, the webpage is in the Chinese language, which makes it inaccessible to readers who do not understand Chinese. Considering the open access nature of this journal, it is preferable to make the info more accessible to everyone by putting it in the supplementary data files.

Thanks for the reviewer's comments. We added the 82 undecapeptides dataset in the Supplementary file detailed in Supplementary Table S1.

c) For molecular docking, the authors used the crystal 1O8A, which is a testes isoform of ACE. Why didn’t the authors use the crystal for the somatic ACE instead, e.g., the crystal 4APJ? After all, the somatic ACE is more broadly distributed in the body when compared with the testis ACE which is restricted to the sperms.

Thanks for the reviewer's comments. We chose the 1O8A crystal structure for molecular docking mainly based on literature reports, which are listed below:

(1) Yin, Z.T.; Yan, R.Y.; Jiang, Y.S.; Feng, S.B.; Sun, H.L.; Sun, J.Y.; Zhao, D.R.; Li, H.H.; Wang, B.W.; Zhang, N. Identification of peptides in Qingke baijiu and evaluation of its angiotensin converting enzyme (ACE) inhibitory activity and stability. Food Chem 2022, 395, 133551.

(2) Li, Z.Y.; He, Y.; He, H.Y.; Zhou, W.Z.; Li, M.R.; Lu, A.M.; Che, T.J.; Shen, S.D. Purification identification and function analysis of ACE inhibitory peptide from Ulva prolifera protein. Food Chem 2023, 401, 134127.

(3) Shao, M.Y.; Wu, H.X.; Wang, B.H.; Zhang, X.; Gao, X.; Jiang, M.Q.; Su, R.H.; Shen, X.R. Identification and characterization of novel ACE inhibitory and antioxidant peptides from Sardina pilchardus hydrolysate. Foods 2023, 12, 2216.

(4) Bhadkaria, A.; Narvekar, D.T.; Nagar, D.P.; Sah, S.P.; Srivastava, N.; Bhagyawant, S.S. Purification, molecular docking and in vivo analyses of novel angiotensin-converting enzyme inhibitory peptides from protein hydrolysate of moth bean (Vigna aconitifolia (Jacq.) Marechal) seeds. Int J Biol Macromol 2023, 230, 123138.

(5) Fu, Y.; Alashi, A.M.; Young, J.F.; Therkildsen, M.; Aluko, R.E. Enzyme inhibition kinetics and molecular interactions of patatin peptides with angiotensin I-converting enzyme and renin. Int J Biol Macromol 2017, 101, 207-213.

(6) Yuan, L.; Sun, L.P.; Zhuang, Y.L. Preparation and identification of novel inhibitory angiotensin-I-converting enzyme peptides from tilapia skin gelatin hydrolysates: inhibition kinetics and molecular docking. Food Funct 2018, 9, 5251-5259.

(7) Zheng, Y.J.; Wang, X.Y.; Guo, M.; Yan, X.T.; Zhuang, Y.L.; Sun, Y.; Li, J.R. Two novel antihypertensive peptides identified in millet bran glutelin-2 hydrolysates: purification, in silico characterization, molecular docking with ace and stability in various food processing conditions. Foods 2022, 11, 1355.

As shown by the NCBI online Protein Blast results, the sequence comparison E-value of 4APJ and 1O8A was zero, so the two sequences were consistent (as shown in the figure below), and the molecular docking using either one of them could illustrate the interaction mechanism of peptide molecules with the ACE receptor.

d) In lines 111-113, the authors mentioned some “critical amino acid residues”. Please cite references which report that those amino acid residues are critical.

Thanks for the reviewer's comments. We added the cite reference, and please check the revised parts of line 117.

e) For molecular docking and dynamics, did the authors perform three replicates too, or just one replicate? Was redocking performed to validate that the docking parameters adopted were reliable? Also, what were the docking parameters used? The information appears not indicated in the manuscript. Please check.

Thanks for the reviewer's comments. Both molecular docking and molecular dynamics simulations were performed with 3 replications to verify the stability of the molecular simulation results. The docking sites in the ACE receptor were 133 active amino acid residues (Figure 1A, line 208), computed and applied by the Site Finder. The score London dG and number of poses were defaulted in MOE. We added the detailed parameters setting information, and please check the revised parts of lines 120-122 and 139-141.

f) For molecular dynamics simulation, 50 ns is rather short. Did the authors attempt to simulate for up to 100 ns, and perform replicates too?

Thanks for the reviewer's comments. We tried a simulation time of 100 ns in the pre-experiment, and the results showed that both could run to stability at 50 ns, so the molecular dynamics simulation time of 50 ns was used for the study. The simulation results show that after 15 ns, RMSD and Rg tend to stabilize, indicating that the peptide molecule can form a stable complex with the ACE receptor.

5. For prediction of anti-ACE potential, the authors searched the BIOPEP-UWM database to see whether their undecapeptides contain previously reported short anti-ACE sequences. While this is often done by others as reported in the literature, it may also restrict the discovery of novel anti-ACE sequences. The authors can consider using various free online anti-ACE peptide prediction servers to screen their sequences instead.

Thanks for the reviewer's comments. We are also trying various free online anti-ACE inhibitory peptide prediction servers, such as DFBP (http://www.cqudfbp.net/), Bioware (http://bioware.ucd.ie/~compass/ Bioware (https://webs. iiitd.edu.in/raghava/ahtpdb/~compass/biowareweb/) to obtain more novel ACE inhibitory peptide sequences. These servers are particularly efficient in predicting novel ACE-inhibiting oligopeptides.

6. Did the authors compare the anti-ACE activity of GQEDYDRLRPL with any well-established anti-ACE agent or antihypertensive drugs?

Thanks for the reviewer's comments. We used the commercial antihypertensive drug Benazepril hydrochloride as a control to compare and evaluate the antihypertensive effect of the screened peptide GQEDYDRLRPL. The results showed that GQEDYDRLRPL exhibited good antihypertensive effects at the same dosage as the commercial drug (recommended dosage, 10 mg/kg body weight) (Figure 6).

7. Discussion can be improved. The first three paragraphs sound like literature and hardly discuss any data/results obtained in this study.

Thanks for the reviewer's comments. We supplemented the discussion section, and please check the revised part of lines 388-390, 400-402, and 415-422.

8. In the section 2.7 (lines 187 – 189), it seems inappropriate to describe MOE 2019 and MD simulation software here as these are not considered to be statistical software.

Thanks for the reviewer's comments. We removed this information and added it to the respective methods sections. Please check the revised part of lines 123 and 142-143.

Reviewer 2 Report

The submitted manuscript presents a comprehensive study on the interaction mechanism and ACE inhibitory activity of long-chain peptides derived from Stropharia rugosoannulata. This work is of significance for the following reasons:

Identification of ACE Inhibitory Peptides: The Angiotensin-Converting Enzyme (ACE) plays a pivotal role in hypertension, and any compounds showing inhibitory activity against this enzyme have profound therapeutic implications. The identification of the undecapeptide GQEDYDRLRPL and its demonstrated activity is a significant advancement in the quest for novel antihypertensive agents.

Natural Source of Inhibitors: The derivation of these peptides from Stropharia rugosoannulata, a fungus, underscores the potential of exploring natural sources for drug development. This avenue can pave the way for more biocompatible and potentially safer therapeutic interventions in contrast to wholly synthetic approaches.

Detailed Mechanistic Insights: Beyond mere identification, this manuscript delves deep into the mechanism of action, offering insights into how the identified peptide binds to and interacts with the ACE receptor. Such knowledge is instrumental for the rational design of more potent and side-effect-minimized drugs.

Robust Experimental Approach: The authors have employed both in vitro and in vivo studies, ensuring a comprehensive evaluation of the peptide's efficacy. This dual-method approach strengthens the reliability and applicability of the findings.

Therapeutic Implications: The in vivo demonstration of the peptide's superior antihypertensive effects, especially at low-gavage administration, suggests that it could potentially be a strong contender as an alternative or supplementary agent to the current ACE inhibitors in the market.

Given the above strengths and the potential impact of this research on the field of antihypertensive drug discovery. The findings presented are both novel and of significant importance, and they undoubtedly contribute to the scientific community's broader understanding of ACE inhibitory agents from natural sources.

English is fine.

Author Response

We express our gratitude to the reviewers for acknowledging our work.